# Approaches Used to Describe, Measure, and Analyze Place of Practice in Dentistry, Medical, Nursing, and Allied Health Rural Graduate Workforce Research in Australia: A Systematic Scoping Review

**DOI:** 10.3390/ijerph19031438

**Published:** 2022-01-27

**Authors:** Hannah Beks, Sandra Walsh, Laura Alston, Martin Jones, Tony Smith, Darryl Maybery, Keith Sutton, Vincent L Versace

**Affiliations:** 1School of Medicine, Deakin University, Geelong 3220, Australia; hannah.beks@deakin.edu.au (H.B.); laura.alston@deakin.edu.au (L.A.); 2Department of Rural Health, University of South Australia, Whyalla 5608, Australia; Sandra.Walsh@unisa.edu.au (S.W.); martin.jones@unisa.edu.au (M.J.); 3Department of Rural Health, University of Newcastle, Taree 2430, Australia; tony.smith@newcastle.edu.au; 4Department of Rural Health & Indigenous Health, Monash University, Warragul 3820, Australia; darryl.maybery@monash.edu (D.M.); keith.sutton@monash.edu (K.S.)

**Keywords:** rural health, health workforce, health occupations, health economics and organizations, allied health occupations, nursing, dentistry, medicine

## Abstract

Redressing the maldistribution of the health workforce in regional, rural, and remote geographical areas is a global issue and crucial to improving the accessibility of primary health care and specialist services. Geographical classification systems are important as they provide an objective and quantifiable measure of access and can have direct policy relevance, yet they are not always consistently applied in rural health research. It is unclear how research focusing on the graduate health workforce in Australia has described, measured, and analyzed place of practice. To examine approaches used, this review systematically scopes Australian rural studies focusing on dentistry, medicine, nursing, and allied health graduates that have included place of practice as an outcome measure. The Joanna Brigg’s Institute Scoping Review Methodology was used to guide the review. Database searches retrieved 1130 unique citations, which were screened, resulting in 62 studies for inclusion. Included studies were observational, with most focusing on the practice locations of medical graduates and predicators of rural practice. Variations in the use of geographical classification approaches to define rurality were identified and included the use of systems that no longer have policy relevance, as well as adaptations of existing systems that make future comparisons between studies challenging. It is recommended that research examining the geographical distribution of the rural health workforce use uniform definitions of rurality that are aligned with current government policy.

## 1. Introduction

Improving the accessibility of health care services for populations residing in regional, rural, and remote geographical areas is a global issue [1]. It is well established that populations residing outside metropolitan areas, even in high-income countries, experience a higher burden of disease and poorer access to health services, when compared with populations residing in metropolitan areas [2,3,4]. Redressing the maldistribution of the health workforce in non-metropolitan areas is integral to improving health care accessibility and outcomes [1,5]. Key to this is the training of health professionals in regional, rural, and remote locations through high-quality education experiences, as supported by national and international evidence [6,7,8]. Research has found that students who have completed an extended period of their undergraduate medical education and training in one or more rural locations are more likely to practice in these settings as graduates [7]. For nursing and allied health students, cumulative rural placements of greater than 20 days have been associated with a greater likelihood of rural practice following graduation [9]. Studies have also found that allied health graduates of rural origin are over twice as likely to practice in a rural or remote workplace, when compared to graduates from metropolitan areas [6].

There is a substantial body of literature examining issues mediating the recruitment and retention of the rural and remote health workforce, with a large proportion of this literature from Australia, Canada, and the United States [8,10]. In Australia, research examining the distribution of the health workforce and issues of recruitment and retention, has tended to dichotomize geographical areas as ‘rural’ or ‘metropolitan’ when presenting findings, consistent with limitations of the broader rural health literature [11,12,13]. International studies have also used the ‘rural’ or ‘non-rural’ dichotomy to illustrate differences in health outcomes for populations residing across geographical areas [3]. In observational research undertaken in Australia, objective measures for stratifying the geographical location of the health workforce have also been used (such as Rural, Remote and Metropolitan Areas (RRMA) [14], the Australian Bureau of Statistics’ (ABS) Australian Statistical Geography Standard—Remoteness Areas (ASGS-RA) or Australian Standard Geographical Classification Remoteness Areas (ASGC-RA)) [6,7].

The importance of using geographical classification systems in research to inform policy and the allocation of resources and understand rural health disparities has been supported by scholars [13,15]. This is particularly important for research undertaken as part of national programs that use geographical classification system to guide program funding priority areas. An example of this is the Rural Health Multidisciplinary Training (RHMT) program [16], which aims to redress the maldistribution of the health workforce. The RHMT program applied the ASGC-RA and ASGS-RA in agreements from 2016–2021 [17] and will use the Modified Monash Model (MMM) from 2022 [18]. Although the use of area-level geographical classification systems is of value when examining the distribution of the health workforce and resources at a national level, and for alignment with prevailing policies, more localized approaches using geographical point data (i.e., a location based upon an x, y coordinate) may be useful. This will advance knowledge of the heterogeneity of distribution of the graduate health workforce across different levels of remoteness. Specifically, this would enable a closer examination of the relationship between the location of training to obtain initial health professional qualification and graduate place of practice. In other research, the importance of place-based approaches to support the retention of the rural and remote health workforce has been emphasized, including facilitating social connection and place integration, and considering other social and professional determinants of rural workforce retention [19]. This phenomenon is yet to be more closely examined in terms of quantifiable geographical measures of location.

The purpose of this review was to scope the literature for objective approaches used to examine the place of practice of dentistry, medical, nursing, and allied health graduates in rurally focused research undertaken in Australia. This is important in understanding how place of practice following graduation is objectively measured, to identify opportunities to improve the consistency of geographical reporting between studies [20] and consider other approaches that could be used. Our initial search of the literature found one scoping review protocol that aimed to examine the study designs and outcomes of allied health and nursing student and graduate tracking studies. That protocol did not specify a focus on the measurement of rural practice outcome measures [21]. Furthermore, it did not seek to examine place of practice for medical graduates. An international systematic review and meta-analysis was also identified which examined the effects of rural pipeline factors, such as rural origin and rural education, on rural practice, but did not explicitly focus on objective measurements of place of practice [22]. Another systematic review examined interventions for health workforce retention in rural and remote areas but did not consider approaches used by studies to describe, measure or analyze place of practice [8]. Rather, this review identified interventions to increase the retention of health workers in rural and remote areas [8].

The research question for this scoping review was as follows:

How is place of practice of graduates described, measured, and analyzed in rural research studies focusing on dentistry, medical, nursing, and allied health graduates in Australia?

The review objectives were to:
[1]Scope Australian rural studies focusing on dental, medical, nursing, and allied health graduates that have included graduate place of practice as an outcome measure and examine objective approaches to analysis; and[2]Examine reported findings and implications of studies, including alignment of approaches used to objectively measure place of practice with relevant state-wide and national policies.


## 2. Materials and Methods

This systematic scoping review examined approaches to describing, measuring, and analyzing place of practice in rural research studies focused on dentistry, medical, nursing, and allied health graduates. The Joanna Briggs Institute’s (JBI) scoping review methodology was used to guide the review [23]. Search terms were developed using Population, Concept and Context (PCC), aligning with a scoping review methodology [23]. The Preferred Reporting Items for Systematic Reviews and Meta-Analysis extension for Scoping Reviews (PRISMA-ScR) [24] checklist was used as a guide for reporting (Appendix A). The review question, objectives, inclusion/exclusion criteria, and search strategies were specified in advance and registered with Open Science Framework (doi:10.17605/OSF.IO/JHFNP).

### 2.1. Search Strategy

The JBI three-step search process guided the development of the search strategy [23]. This involved a preliminary search undertaken in Ovid MEDLINE and CINAHL using keywords. A tailored search for each information source was then developed, using keywords. A combination of Boolean operators, truncations, and Medical Subject Headings or EMTREE headings, were used to form search strings (Appendix A). Two librarians with expertise in developing search strategies for databases, reviewed the searches. Reference lists of included studies were reviewed for additional studies.

Databases searched included Ovid MEDLINE, CINAHL (EBSCOhost), PsycInfo (EBSCOhost), and Embase (Elsevier). Google Scholar was also searched to capture any additional studies. A search for grey literature was not undertaken due to the inclusion specifying only peer-reviewed research. Database searches were completed on 3 September 2021.

### 2.2. Inclusion and Exclusion Criteria

Literature was screened according to the inclusion and exclusion criteria shown in Table 1. Peer-reviewed observational or experimental research studies were included that focused on dentistry, medical, nursing, and allied health graduates, and which used place of practice as a primary or secondary outcome measure. Graduates were defined as those who had completed training to obtain an initial health professional qualification and excluded those who completed post-graduate training and specialist programs. Studies had to specify the year(s) of graduation or enrolment. Literature published after 1 January 2010 were included, to align with a greater global focus on the need for targeted strategies to recruit health professionals to rural and remote areas [1]. Mixed-methods studies with an embedded observational or experimental study component were included. Only studies published in English were included due to resource constraints.

### 2.3. Study Selection and Data Extraction

Citations were imported into Covidence (Veritas Health Innovation, Melbourne, Australia) for screening. Titles and abstracts were screened independently by at least two reviewers, with conflicts resolved through discussion with a third reviewer. Full text review and data extraction was then undertaken. For articles reviewed at full text screening not meeting the inclusion criteria, reasons for exclusion were provided (Appendix A). Reference lists of included studies were screened for additional studies. Data extraction was tabulated using the following headings: author, year, study design, participant sample and demographics, outcome measures, approach to the measurement, description, and analysis of place of practice, findings, implications, and study limitations. Findings were synthesized using a descriptive approach as informed by the review question and objectives [23]. A quality assessment of included studies was not undertaken, as is conventional for scoping reviews [25].

## 3. Results

Of the 103 citations eligible for full text screening, 62 studies met the inclusion criteria. No additional citations were identified through a review of the reference lists of included studies and reasons for exclusions are shown in Figure 1. PRISMA Flow Diagram and Appendix A.

### 3.1. Characteristics of Included Studies

A summary of included studies included is provided in Appendix A. No experimental or quasi-experimental studies were reported. Of observational studies described, over two thirds were cohort designs (*n* = 44, 71%), nearly a quarter cross-sectional (*n* = 15, 24%), and the remainder mixed methods (*n* = 3, 5%) with an embedded cohort study design. The majority of studies tracked medical graduate outcomes (*n* = 51, 82%). Studies of other professions (*n* = 11, 18%) were limited to dentistry (*n* = 4), physiotherapy (*n* = 1), optometry (*n* = 1), pharmacy (*n* = 1), nursing (*n* = 1), multiple allied health professions (*n* = 1), and nursing and multiple allied health professions (*n* = 2). Most studies focused on graduates from a single university (*n* = 51, 82%) (including one study with a local health district focus) with a small number of collaborative studies undertaken by more than one university (*n* = 3). Of other studies, five had a national focus and three had a statewide focus.

Of the university studies, most employed a cohort approach (*n* = 38, 70%) with repeated surveys and/or data linkage (including university data and Australian Health Practitioner Regulation Agency (AHPRA) data). Nearly a quarter were cross-sectional studies involving surveys, university and/or AHPRA data (*n* = 13, 24%), and the remaining three (6%) studies utilized a mixed methods design with an embedded cohort study. Most studies were undertaken within only one jurisdiction, including Western Australia (*n* = 17, 31%), Queensland (*n* = 15, 28%) (of which one included an international cohort [26]), Victoria (*n* = 7, 13%), New South Wales (*n* = 6, 11%), South Australia (*n* = 3, 5%), Tasmania (*n* = 3, 6%), and the Australian Capital Territory (ACT) (*n* = 1, 2%). Two studies (4%) were conducted in two or more Australian states/territories.

### 3.2. Approaches to Measuring, Analyzing, and Describing Place of Practice

Similar methods were used to collect place of practice data (e.g., suburb, postal areas (POAs)), which included either self-reporting via survey (undertaken over the phone or online), or data linkage methods using AHPRA data, and/or other open access online sources (e.g., Google, yellow pages (https://www.yellowpages.com.au/) accessed on 20 September 2021). However, a diverse range of approaches was used to analyze and describe place of practice. Most studies reported using an established geographical classification system to analyze and describe place of practice (*n* = 54, 87%) and to provide an objective measure of rurality or provided information around an alternative approach used to define rurality or analyze/describe place of practice (e.g., use of Index of Relative Socio-economic Advantage and Disadvantage (IRSAD) [27], or other criteria) (*n* = 6, 10%). One study used both the Accessibility/Remoteness Index of Australia (ARIA) and Socio-Economic Indexes for Areas (SEIFA) to analyze and describe practice locations [28]. Two studies used multiple established geographical classification systems [29,30], while two other studies did not clearly specify how rurality was defined [31,32].

Table 2 illustrates that ASGC-RA was the most frequently used geographical classification system to analyze and describe place of practice (*n* = 32). In order of frequency, the other classification systems used were MMM (*n* = 13), RRMA system (*n* = 6), ASGS-RA (*n* = 4), and the Accessibility/Remoteness Index of Australia (ARIA) (*n* = 1). Two studies used both ASGC-RA and MMM to analyze and describe place of practice [29,30].

Four studies used an alternative approach to define rural, one used population sizes to classify geographical areas but did not cite an established classification system [86]. Another used proximity from a city post office to define rural [87] and three used State/Territory borders or other geographical boundaries to analyze and describe place of practice [26,88,89]. A single study classified place of practice using ISRAD to provide an indication of where medical graduates were practicing in terms of socio-economic deciles [27].

Approaches to distinguishing rural from metropolitan practice varied between studies, and generally involved merging geographical classifications into broader categories. Studies using the RRMA usually classified RRMA1-2 as metropolitan and RRMA3-5 as rural, although two studies classified RRMA 3-7 as rural. Twenty-two of the 32 studies using the ASGC-RA classification dichotomized rural or non-metropolitan as RA2-5 and metropolitan or urban as RA1, while a single study dichotomized RA1-3 as non-remote and RA4-5 as remote [53]. Two of the four studies using the ASGS-RA dichotomized major cities as RA1 and rural as RA2-5. Studies using the MMM varied in their approaches of classifying rural, with six studies classifying rural as MMM2-7.

### 3.3. Findings and Implications

Although there were variations in the approaches used to analyze and describe place of practice, and define rural practice, similarities in findings and implications were evident across the included studies. These are synthesized below according to profession.

#### 3.3.1. Dentistry

There were four studies with a focus on the practice locations of graduate dentists (two cohort studies and two cross-sectional studies) with varying approaches to analyzing and describing place of practice. A retrospective cohort study, from a single university, found that 51% of dentistry graduates practiced in high socioeconomic areas and areas that were classified as highly accessible using the ARIA classification system, while fewer than 12% were practicing in low socioeconomic areas [28]. Another cohort study found that a higher proportion of graduates who had participated in a rural placement program were practicing in a rural location, when compared to those who had not undertaken a rural placement [35]. Those findings were expanded in an evaluation of a Rural Clinical Placement Program (RCPP) offered to final-year dentistry students by the same university, which identified that the odds of working in a rural location were higher for graduates who participated in the RCPP compared to those who did not (OR = 1.83, 95% CI 1.00, 3.56) [34]. Similarly, a cross-sectional study across multiple dentistry programs from metropolitan and non-metropolitan universities found that dentistry graduates who studied in a rurally focused program were statistically more likely to be practicing in MMM2-7 locations compared to those who had studied in a metropolitan-based program [84].

#### 3.3.2. Medicine

Of 51 studies examining the place of practice of medical graduates, 40 were either cohort studies, longitudinal cohort studies, or mixed methods studies with an embedded cohort study. Of these studies, different models of rural clinical placements and training were supported. This included the value of longitudinal integrated clerkships (LIC) in rural areas as a strategy to recruit graduates to rural areas [37], with graduates who had completed a rural LIC 5.6 times (95% CI 2.8–11.2) more likely to work in smaller regional or rural towns when compared to other graduates [75]. Other studies identified that graduates who had participated in a rural clinical school (RCS) during their training, were more likely to undertake rural and remote work in years following graduation [15,30,31,36,38,42,47,48,52,57,59,60,61]. However, it was identified that this rural work was not always sustained [42] and that graduates had limited prior contact with the rural areas in which they were working [50]. Other models of rural training included a partnership between a university and a health service to deliver a rural training program, with research finding that graduates who had participated in the program were more likely to remain local for an internship, as well as following an internship [44]. Overall, rural training was supported as a predictor of rural practice [65,66,67,78,81,89].

Other predicators of rural practice identified in cohort studies of medical graduates included: rural preference prior to medical school or rural practice intention [43,51,58]; being of rural origin or background [11,39,48,51,52,58,60,63,78]; having extended periods of rural clinical training [56,77,79,80]; completing schooling and/or training in the same rural region [77,88]; being a graduate with prior tertiary education [49]; having a partner with a rural background [60]; being single [60]; being of a mature age at entry [62]; and having a bonded scholarship [60]. However, rural background as a predictor of rural practice was found to diminish in the years following graduation [58]. One study also found that graduates from regions with lower socio-economic deciles at the commencement of their studies were more likely to practice in lower socio-economic regions in the first five years after graduating (OR 2.05, 95% CI, 1.71) [27].

Studies identified a range of challenges related to classifying the geographical distribution of medical workforce, which were as follows: uptake of rural internships by domestic graduates being insufficient to meet rural workforce demands [76]; movement of graduates who were international fee-paying students to metropolitan areas after the first few years of practice [82]; and the lack of availability of specialty training in non-metropolitan areas [57]. The sensitivity of AHPRA principal place of practice data was also examined in one study, which found that the most accurate method of tracking medical graduates longitudinally was through personal contact [32], as long as that remains possible.

Of the 11 cross-sectional studies examining the place of practice of medical graduates, studies broadly described the location of graduates and proportion of graduates practicing in rural areas with the use of an established geographical classification system [26,29,46,54,55,64,74,86]. One study adopted an alternative approach and examined the proportion of graduates practicing in the defined geographical footprint of the university [26]. That study found that 38% of graduates practiced within that footprint and provided support for rural and regional training as a strategy to increase medical graduates in those areas [26]. One cross-sectional study examining RCS graduates found a higher proportion of graduates practicing in rural areas after their fifth postgraduate year compared with their first five years of practice [46]. Predictors of rural practice were also identified, including: having a rural background [45,74,83]; participating in an extended RCS experience [29,83]; having the opportunity to participate in rural general training [64]; and having a bonded rural scholarship [45]. Predictors of remote practice were completing rural generalist training, undertaking an outer-regional or remote internship, and being female [53].

#### 3.3.3. Nursing and Allied Health

Three studies examined the practice locations of nursing graduates, two of which also included allied health graduates. In the study that examined nursing graduates only, a higher proportion of graduate nurses who had studied in a rural-based program were practicing in rural locations compared to those who did not study in a rural-based program [87]. Of the two studies that included both nursing and allied health graduates, predictors of rural practice were identified. Entering rural practice in the first year after graduation was a predictor of long-term rural practice for graduates 15 to 17 years after their rural placement [69], while being of rural origin and having more rural placement days were also predictors of rural graduate practice [9].

Of the remaining four studies that focused on allied health professions, which were pharmacy (*n* = 1), physiotherapy (*n* = 1), optometry (*n* = 1), and multiple allied health disciplines (*n* = 1), two studies identified rural background as a predictor of graduate rural practice [68,85]. Other research identified the challenges of retaining physiotherapy graduates in rural areas [71] and the high proportion of graduate optometrists practicing in major cities of Australia [72].

## 4. Discussion

This review identified considerable variability in the application of geographical approaches to examine place of practice in rural health graduate research in Australia. A greater focus on the place of practice of medical graduates was also identified which reflects the maturity of medical tracking relative to nursing and allied health professions [90]. Findings illustrate application differences in the use of geographical classification systems and definitions of rurality across health professions over the past 10 years. The use of redundant geographical classification systems relative to year of publication and study period (e.g., RMA which uses 1991 population counts) to analyze and describe practice locations was unexpected. Using geographical classification systems of relevance to policy is important to facilitating the translation of research findings into decision-making, and evaluating the impacts of government programs and funding models [15,20]. In the Australian policy context, government programs (e.g., RHMT program [16], pharmacy rural support programs [91]) are transitioning to the MMM to inform program priorities and funding allocation [18]. With the transition to the use of the MMM at a national policy level, it would be appropriate for researchers to also transition to using the MMM when examining the geographical distribution of the health workforce, including the practice locations of health graduates [92]. However, there are limitations to applying geographical classification systems that use population size and geographical remoteness as proxy measures of access for health service planning and resource allocation [93]. Other tools of measuring access have been developed, including the Index of Access which measures access to primary health care services and can be used to inform workforce planning [93]. The need for specific access models to be developed and updated to assist with the efficient deployment of resources and targeted service provision has also been supported by other research [94].

Definitions of rurality were found to vary across the studies included in this review. Differences largely related to the use of geographical classification systems and the approach used to dichotomize ‘rural’ and ‘non-rural’, which neglects the large variation in characteristics of place. For example, it was common for studies that used the ASGC-RA or revised ASGS-RA to describe those locations that fitted the RA1 category as metropolitan or urban and RA2-5 as rural or non-metropolitan. Globally, defining rurality has been problematic for policymakers and researchers [95]. In Australia, the ABS acknowledges that remoteness is dynamic as populations move across geographical areas and infrastructure is developed [70]. In navigating the complexities of defining rurality, Bennett and colleagues (2019) advise that researchers should provide a specific definition of rurality in their work which aligns with either the research purpose, intended audience and/or funding source, and the approach used by previous research or data collection methods [95]. Inconsistencies in the definition of rurality in research and use of different approaches to describe and analyze place of practice is problematic, not least because it makes it difficult to compare and synthesize the findings of studies [20]. In the context of this review, studies largely reported the research activity of a single university examining their own graduates, with few studies having a coordinated approach or joint endeavor of multiple universities.

Another key finding is that almost three quarters of past research focused on medical graduates. The next largest health profession studied was dentistry with four studies, leaving the other health professions largely not investigated. Few studies examined nurses who, numerically, make up the majority of the rural health workforce and even fewer studies explored allied health professions. The paucity of non-medical workforce research when combined with the different conceptualization of rurality highlights some key knowledge gaps which need to be addressed, noting the recent establishment of the Nursing and Allied Health Graduate Outcome Tracking (NAHGOT) study that is compiling a multi-institutional, multidisciplinary, longitudinal database to address this need [96]. In Australia, and internationally, there has historically been a significant focus on the medical workforce at a policy level, including training of students from rural backgrounds to redress rural and remote health workforce needs, which is evidenced by the studies retrieved [97]. Other reviews have also found rural and remote health workforce studies to be observational and predominantly medical workforce-focused [8]. However, research designed to address these knowledge gaps is in progress. The NAHGOT study has the potential to address some of the knowledge gaps regarding the place of practice of nursing and allied health graduates [96].

Although there is a need for improved consistency and more collaborative research between universities to examine the graduate health workforce, there is value in adopting more localized approaches. For example, using area level geographical data to examining the practice locations of health graduates, across all states of Australia. This is particularly useful for determining the effectiveness of clinical training delivered in rural areas in meeting the demands of the local health workforce. Such an approach was used by Wolley et al. who analyzed the proportion of graduates practicing within the university geographical footprint to examine how the university was contributing to the medical workforce supply in the region [26]. This approach would also be useful to evaluating the effectiveness of evidence-based strategies implemented by universities and health organizations to target students from rural backgrounds, implement distributed clinical training models, and provide support for ongoing training in years following graduation [8]. Indeed, moving beyond area-level geographical definitions of rurality (e.g., RA2-RA5, MM2-MM7, or a geographical footprint) and to locality data (e.g., town or suburb) to track graduate movement offers much promise to further understanding graduate place of practice. For example, at scale this approach would allow investigations into the influence of end-to-end training pathways and return to region, the results of which will potentially allow universities to better target recruitment strategies to select those entrants most likely to practice rurally upon completion of their training.

### Limitations

Due to the review question and defined scope of this review, only peer-reviewed studies were included. The potential for publication bias is a limitation of this review due to the exclusion of grey literature studies, including health workforce evaluation reports. Including health reports and graduate data (unpublished) may have helped to highlight more consistent use of classification systems and data collection methods, particularly within states; however, the aim of this review was to investigate peer reviewed literature. Furthermore, given the particular nature of the geographical classification systems investigated, the focus of the review was limited to the Australian context, as per the review question and objectives. Future research could also examine international approaches to measuring, analyzing, and describing place of practice in graduate health workforce research in other high-income countries that also have a maldistribution of the health workforce, such as Canada and the United States. This would serve to expand the range of geographical classification systems and allow international comparisons. Additionally, this study focused on how published studies described, measured, and analyzed place of practice in the context of standardized geographical systems. Therefore, we did not analyze the broader historical context of place of practice or social interactions that influence the distribution of health workers. These are important priorities for future research focus in rural health workforce research.

## 5. Conclusions

Variations in the use of geographical classification systems and definitions of rurality are evident in the literature. It is recommended that future research examining the geographical distribution of the rural health workforce use definitions of rurality that are aligned with current government policy and to facilitate longitudinal graduate tracking. Consideration should also be given to the use of local area level geographical analysis to examine more closely the relationship between training programs and the actual practice location of graduates. While there is a large volume of literature relating to medical graduates, further research is required into the place of practice of dentistry, nursing, and allied health graduates in Australia, as rural health services and communities experience critical workforce deficiencies across these disciplines. In addition, to better inform health workforce planning and education strategies, there is a need for coordinated efforts between universities, health service providers and governments to develop the evidence-base around the health workforce in both metropolitan and non-metropolitan areas.

## Figures and Tables

**Figure 1 ijerph-19-01438-f001:**
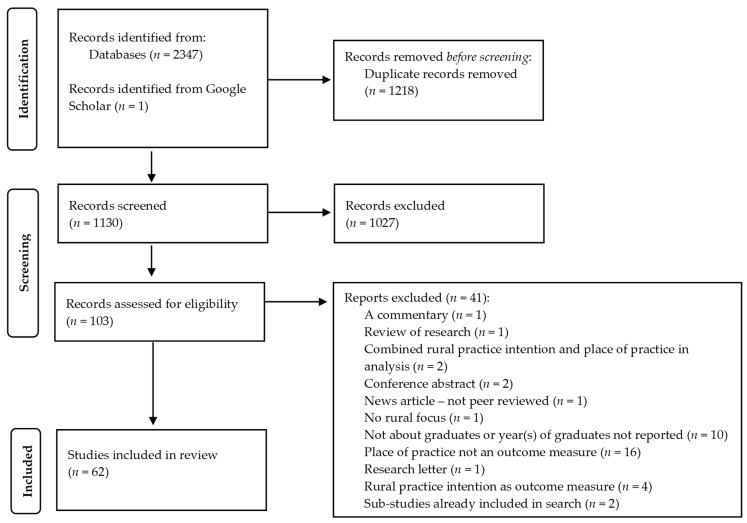
Preferred Reporting Items for Systematic Reviews and Meta-Analysis (PRISMA) Flow Diagram. From: Page, M.J.; McKenzie, J.E.; Bossuyt, P.M.; Boutron, I.; Hoffmann, T.C.; Mulrow, C.D. The PRISMA 2020 statement: an updated guideline for reporting systematic reviews. *BMJ* **2021**, *372*, n71. https://doi.org/10.1136/bmj.n71.

**Table 1 ijerph-19-01438-t001:** Inclusion and exclusion criteria.

Parameter	Inclusion	Exclusion
Population	Studies focused on dentistry, medical, nursing, and allied health graduates. Allied health includes, but is not limited to, speech pathology, occupational therapy, podiatry, audiology, exercise physiology, medical imaging, psychology, physiotherapy, pharmacy, social work, chiropractors, and dietitians.	Non-health professions (e.g., teaching)
Concept	Peer-reviewed experimental, observational research studies (e.g., longitudinal studies, case-control studies, cohort studies), and mixed-method studies including place of practice as a primary or secondary outcome measure including, but not limited to, principal place of practice location following graduation.	Study protocols and non-peer reviewed studies including grey literature such as opinion-based articles and evaluation reports. Studies including intention for rural practice as an outcome measure.
Context	Studies with a focus on the Australian setting, published since 1 January 2010.	Not published in English. Reference only to non-Australian settings.

**Table 2 ijerph-19-01438-t002:** Geographical classification systems used.

Geographical Classification System	Included Studies by Occupational Category
Rural, Remote and Metropolitan Areas Classification 1994 [33]; Divides Statistical Local Areas (SLAs) into seven population size categories: Capital city; Other metropolitan (pop. ≥ 100,000); Large rural (pop 25,000–99,999); Small rural (10,000–24,999); Other rural area (<10,000); Remote zones (>5000); and Other remote (<5000).	Dentistry: Johnson, Wright and Foster 2019 [34], Johnson and Blinkhorn 2013 [35]
	Medicine: Clark et al., 2013 [36], Kitchener et al., 2015 [37], Playford and Cheong 2012 [38], Strasser et al., 2010 [39]
Accessibility/Remoteness Index of Australia 1999 [40]; Superseded RRMA and calculates accessibility to service centres based on road distances. Values are scored in five categories: Highly Accessible (ARIA score 0–1.84); Accessible (ARIA score > 1.84–3.51); Moderately Accessible (ARIA score > 3.51–5.80); Remote (ARIA score > 5.80–9.08); and Very Remote (ARIA score > 9.08–12).	Dentistry: Gurbuxani, Kruger and Tennant 2012 [28]
Australian Standard Geographical Classification Remoteness Areas [41]; based on ARIA, the ASGC-RA was implemented by the ABS using data from the 2001 census and 2006 census, and has five categories based on ARIA, and a sixth category (Migratory): Major Cities of Australia (ARIA 0–0.2), Inner Regional Australia (ARIA > 0.2 and ≤2.4), Outer Regional Australia (ARIA > 2.4 and ≤5.92), Remote Australia (ARIA > 5.92 and ≤10.53), Very Remote Australia (ARIA > 10.53), and Migratory (off-shore, shipping, migratory).	Medicine: Gupta et al., 2019 [42], Herd et al., 2017 [43], Kitchener 2021 [44], Kwan et al., 2017 [45], Mc Grail et al., 2017 [11], Moore et al., 2018 [46], Playford and Puddey 2017 [47], Playford et al., 2017 [48], Playford et al., 2019 [49], Playford, Burkitt and Atkinson 2019 [50], Playford, Ngo and Puddey 2021 [51], Playford, Ngo, Atkinson and Puddey 2019 [52], Woolley, Gupta and Bellei 2017 [53], Woolley, Gupta and Larkins 2018 [54], Lewis et al., 2016 [55], Playford, Ng and Burkitt 2016 [56], Shires et al., 2015 [30], Eley et al., 2012 [57], Hogenbirk et al., 2015 [58], Jamar, Newbury and Mills 2014 [59], Kondalsamy-Chennakesavan et al., 2015 [60], Playford et al., 2014 [61], Playford et al., 2015 [7], Puddey et al., 2015 [62], Ray, Woolley and Sen Gupta 2015 [63], Schauer, Woolley and Sen Gupta [64], Sen Gupta et al., 2013 [65], Sen Gupta et al., 2014 [66], Woolley et al., 2014 [67], McGirr et al., 2019 [29]
	Allied health: Brown et al., 2017 [68]
	Nursing and allied health: Playford, Moran and Thompson 2020 [69]
Australian Statistical Geography Standard Remoteness Areas [70]; replacing the ASGC-RA in 2011 and implemented by the ABS, the ASGS-RA classifies Australia into five categories of remoteness using ARIA+: Major Cities of Australia (ARIA+ 0–0.2), Inner Regional Australia (ARIA+ > 0.2 and ≤2.4), Outer Regional Australia (ARIA+ > 2.4 and ≤5.92), Remote Australia (ARIA+ > 5.92 and ≤10.53), Very Remote Australia (ARIA+ > 10.53).	Physiotherapy: Bacopanos and Edgar 2016 [71]
	Optometry: Duffy et al., 2021 [72]
	Nursing and allied health: Sutton et al., 2021 [73]
	Medicine: Fuller, Beattie and Versace 2021 [74]
Modified Monash Model [18]; developed using the ASGS-RA, the MMM was introduced by the Australian Department of Health in 2015 (based upon 2011 census data) to inform policy targeting incentive payments for doctors practicing in rural areas. The most recent iteration is MMM 2019 and is based upon 2016 census data. The MMM includes seven categories: MM1—Metropolitan areas (ASGS-RA1), MM2—Regional areas (ASGS-RA2 and ASGS-RA3, within a 20 km drive of a town with <50,000 residents), MM3—Large rural towns (ASGS-RA2 and ASGS-RA3—areas that are not MM2 or within a 15 km drive of a town between 15,000 to 50,000 residents), MM4—Medium rural towns (ASGS-RA2 and ASGS-RA3 areas that are not MM2 or MM3 and are within a 10 km drive of a town between 5000 to 15,000 residents), MM5—Small rural towns (all remaining ASGS-RA2 and ASGS-RA3 towns), MM6—Remote communities (ASGS-RA4 and remote islands less than 5 km offshore), and MM7—Very remote communities (ASGS-RA5).	Medicine: Campbell et al., 2019 [75], McGrail et al., 2020 [76], McGrail, O’Sullivan and Russell 2018 [77], O’Sullivan and McGrail 2020 [78], O’Sullivan et al., 2018 [79], O’Sullivan et al., 2019 [80], Walker et al., 2021 [81], Cheek et al., 2017 [82], McGirr et al., 2019 [29], May, Brown and Burrows 2018 [83], Shires et al., 2015 [30]
	Dentistry: Tchia et al., 2019 [84]
	Pharmacy: Drovandi et al., 2020 [85]

## Data Availability

No new data was created for this study, rather information was obtained from included studies. Data sharing is not applicable in this article.

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
