# Peer review of "Approaches Used to Describe, Measure, and Analyze Place of Practice in Dentistry, Medical, Nursing, and Allied Health Rural Graduate Workforce Research in Australia: A Systematic Scoping Review"

_ijerph, 2022, doi:10.3390/ijerph19031438_

Round 1
Reviewer 1 Report
Thank you for the opportunity to review your paper which scoped the literature for objective approaches used to examine the place of practice of dentistry, medical, nursing, and allied health graduates in rurally focused research undertaken in Australia. This review is timely needed and makes a significant contribution to the field. The paper is nicely written and well organized. The methods used are scientifically sound and comprehensively described. Results are interesting and useful. Recommendations for future research are appropriate.
Author Response
Thank you for your comments and for reviewing our paper.
Reviewer 2 Report
The review paper explores the place of practice among different health professionals in rural Australia so as to improve health care access and outcomes.
The paper draws from a substantial body of literature from studies conducted in Australia, Canada and the United States (Line 57) which have huge populations of Aborigine/ Indigenous people. However, the paper fails to broadly provide a historical context that highlights health inequities that feed into the place of practice debate, especially for health professionals in rural areas. Furthermore, I find that as a health geographer, the review paper reduces the “place of practice” to more of location rather than the social interactions that influence the distribution of health workers.
Author Response
We agree with the reviewer that the inclusion of the boarder historical context of indigenous populations across the countries included in this review is a limitation and is beyond the scope of this paper. In using place of practice, we sought to look at measures that attempt to quantify location, and agree that it is also a limitation that we have not investigated the social interactions that influence the distribution of health workers. We have now included the following addition to our limitations “Additionally, this study focussed on how published studies described, measured, and analyzed place of practice in the context of standardised geographical systems. We therefore did not analyse the broader historical context of place of practice or social interactions that influence the distribution of health workers. These are important priorities for future research focus in rural health workforce research.
Reviewer 3 Report
This review clearly examines place of practice measures and analysis and provides high level discussion of how this information can be utilised to improve access to healthcare in remote, rural and regional areas within Australia. Overall, a valuable health literature review across many contexts. There are implications for the health workforce, health education sector and policy both locally and nationally from the findings of your study. A well written review on a timely issue. Some minor comments below.
Can remove any key words that are already in the title.
The introduction is fairly well written although it would benefit from including a definition of place of practice (Lines 91-92) - this would help to provide a clearer understanding of what is being "measured" and "analyzed". (e.g. PoP is the practice location of graduates). The references utilised a relevant and mostly recent and help to highlight the current issues with workforce tracking in health. Lines 96-109 dont belong in the introduction, they should be part of the strategy (method).
The methods section is clear and demonstrates an appropriate, validated strategy for scoping literature. The key terms are well established and there is evidence of peer review which is always valuable.
The results are well presented and demonstrate search rigour. Line 195/196/207 - capitalize AHPRA abbreviation. Line 293 formatting (space missing)
The discussion is well written and show high level thinking in terms of what the results mean. The discussion around the inflated amount of literature in the medical space is particularly interesting, especially in terms of the limitation of not including any grey literature or non-published reports. Including health reports and graduate data (unpublished) may have helped to highlight more consistent use of classification systems and data collection methods, particularly within states.
You have highlighted some very important conclusions for ongoing student/graduate tracking and potential policy implications nationally and locally.
Some minor referencing issues (e.g. #38, #22
Author Response
- We have now clarified in our research introduction research questions that we are investigating the place of practice of graduates. Lines 96-109 have been corrected as this sentence was describing methods used in an initial search that informed this review. This is confusing to the reader and has now been refined.
- Thank you for identifying this. We have edited this error. (Reference in manuscript: lines 195, 196 and 207, 297.)
- We agree with the reviewer that including health reports and graduate data (unpublished) may have helped to highlight more consistent use of classification systems and data collection methods, particularly within states. We have included this in our limitations to say ‘Including health reports and graduate data (unpublished) may have helped to highlight more consistent use of classification systems and data collection methods, particularly within states, however the aim of this review was to investigate peer reviewed literature’. (Reference in manuscript: lines 417-420.)
- We have reviewed the reference list to ensure there are no further errors.
Round 2
Reviewer 2 Report
The authors have sufficiently addressed the comments. I have no further reservations.